# Glycosylation Analysis of Feline Small Intestine Following *Toxoplasma gondii* Infection

**DOI:** 10.3390/ani12202858

**Published:** 2022-10-20

**Authors:** Bintao Zhai, Shichen Xie, Junjie Peng, Yanhua Qiu, Yang Liu, Xingquan Zhu, Junjun He, Jiyu Zhang

**Affiliations:** 1Key Laboratory of Veterinary Pharmaceutical Development, Lanzhou Institute of Husbandry and Pharmaceutical Sciences, Chinese Academy of Agricultural Sciences, Ministry of Agriculture, Lanzhou 730050, China; 2State Key Laboratory of Veterinary Etiological Biology, Key Laboratory of Veterinary Parasitology of Gansu Province, Lanzhou Veterinary Research Institute, Chinese Academy of Agricultural Sciences, Lanzhou 730046, China; 3College of Veterinary Medicine, Shanxi Agricultural University, Jinzhong 030801, China; 4Research Center for Parasites & Vectors, College of Veterinary Medicine, Hunan Agricultural University, Changsha 410128, China; 5College of Veterinary Medicine, Northwest A&F University, Xianyang 712100, China; 6College of Life Science, Ningxia University, Yinchuan 750021, China; 7Key Laboratory of Veterinary Public Health of Yunnan Province, College of Veterinary Medicine, Yunnan Agricultural University, Kunming 650201, China

**Keywords:** *Toxoplasma gondii*, oocysts, glycosylation, *N*-glycosylation

## Abstract

**Simple Summary:**

*Toxoplasma gondii* has a serious impact on public health and the economic development of animal husbandry. Glycosylation, especially *N*-glycosylation, the pattern modification of proteins, is closely related to the biological functions of proteins, and our study used it to analyze glycosylation alterations in the small intestine of cats infected with *T. gondii*. The results of the present study showed that 56 glycosylated peptides were upregulated and 37 glycosylated peptides were downregulated. Additionally, we also identified eight *N*-glycosylated proteins of *T. gondii* including eight *N*-glycopeptides and eight *N*-glycosylation sites. Moreover, the protein eEF2 and its corresponding peptide sequence were identified, with GO terms (i.e., cellular process and metabolic process, cell and cell part, and catalytic activity) that were significantly enriched in the *T. gondii* MAPK pathway. In addition, the Clusters of Orthologous Groups of proteins (COG) function prediction results showed that posttranslational modification, protein turnover, and chaperones (11%) had the highest enrichment for *T. gondii*. The host proteins ICAM-1 and PPT1 and the endoplasmic reticulum stress pathway may play an important role in the glycosylation of *T. gondii*-infected hosts. Our study may provide a new target for *T. gondii* detection to prevent the spread of *T. gondii* oocysts in the future.

**Abstract:**

*Toxoplasma gondii* (*T. gondii*) is responsible for severe human and livestock diseases, huge economic losses, and adversely affects the health of the public and the development of animal husbandry. Glycosylation is a common posttranslational modification of proteins in eukaryotes, and *N*-glycosylation is closely related to the biological functions of proteins. However, glycosylation alterations in the feline small intestine following *T. gondii* infection have not been reported. In this study, the experimental group was intragastrically challenged with 600 brain cysts of the Prugniuad (Pru) strain that were collected from infected mice. The cats’ intestinal epithelial tissues were harvested at 10 days post-infection and then sent for protein glycosylation analysis. High-performance liquid chromatography coupled to tandem mass spectrometry was used to analyze the glycosylation alterations in the small intestine of cats infected with *T. gondii*. The results of the present study showed that 56 glycosylated peptides were upregulated and 37 glycosylated peptides were downregulated in the feline small intestine infected by *T. gondii*. Additionally, we also identified eight *N*-glycosylated proteins of *T. gondii* including eight *N*-glycopeptides and eight *N*-glycosylation sites. The protein A0A086JND6_TOXGO (eEF2) and its corresponding peptide sequence were identified in *T. gondii* infection. Some special GO terms (i.e., cellular process and metabolic process, cell and cell part, and catalytic activity) were significantly enriched, and the Clusters of Orthologous Groups of proteins (COG) function prediction results showed that posttranslational modification, protein turnover, and chaperones (11%) had the highest enrichment for *T. gondii*. Interestingly, eEF2, a protein of *T. gondii*, is also involved in the significantly enriched *T. gondii* MAPK pathway. The host proteins ICAM-1 and PPT1 and the endoplasmic reticulum stress pathway may play an important role in the glycosylation of *Toxoplasma*-infected hosts. This is the first report showing that *T. gondii* oocysts can undergo *N*-glycosylation in the definitive host and that eEF2 is involved, which may provide a new target for *T. gondii* detection to prevent the spread of *T. gondii* oocysts in the future.

## 1. Introduction

*Toxoplasma gondii* is a strict intracellular parasite that can infect all warm-blooded vertebrates including humans and birds. Generally, infection with *T. gondii* is asymptomatic in healthy individuals but fatal for individuals with immunodeficiency [1]. For AIDS patients, *T. gondii* infection causes secondary encephalitis, pneumonia, and disseminated infection [2]; for pregnant women, the situation is more serious, and *T. gondii* infection can lead to premature birth, miscarriage, deformity, and even stillbirth [3,4]. Although *T. gondii* has a wide range of intermediate hosts, it only carries out sexual reproduction in felids [5]. Once cats are infected with *T. gondii*, bradyzoites settle within enterocytes, undergo sexual development, and form unsporulated oocysts [6]. The unsporulated oocysts are shed through the feces and then sporulated into infectious oocysts [7]. The discharged oocysts can be scattered on land, grass, water, or anywhere [8]. Contaminated water/grass/land, if eaten by livestock, can cause *T. gondii* infection in humans or companion animals [9]. *T. gondii* oocysts are a major public health problem.

Glycosylation is the most extensive and functional posttranslational modification (PTM) of proteins. Approximately 50% of proteins in eukaryotes undergo *N*-glycosylation modification and *O*-glycan [10]. The *N*-glycosylation of proteins is closely related to the occurrence and development of diseases such as diabetes, immune system diseases, nervous system diseases, and cancer [11,12]. Protein *N*-glycosylation plays a key role in numerous eukaryotic biological processes including molecular interactions, signal transduction, cell adhesion, endocytosis, protein stability, and other relevant biological functions [13,14]. In the biosynthesis of *N*-glycans, under the action of glycosidase, *N*-acetylglucosamine (GlcNAc) and mannose (Man) molecules are associated with asparagine (Asn) residues to form the basic structure of *N*-glycans. Galactose (Gal), *N*-acetylneuraminic acid (Neu5Ac), GalNAc, and other molecules participate in the process of sugar chain formation [10,15]. Asparagine-linked glycan (*N*-glycan)-dependent quality control (QC) systems for protein folding and endoplasmic reticulum (ER)-related degradation exist in most eukaryotes; however, in Giardia and some other protists, truncated *N*-glycans have been lost [16,17]. Plasmodium falciparum and *T. gondii* have also been found in lack proteins, which are involved in the *N*-glycan-dependent QC of protein folding [16]. Because the *N*-glycan structure of *Plasmodium falciparum* and *T. gondii* is controversial [18], and the *T. gondii N*-glycosylation function is critical for both parasite motility and host cell invasion [19], we were therefore very interested in whether *N*-glycosylation also occurs in the definitive host infection in *T. gondii*.

Although some articles highlight that the *N*-glycosylation of proteins plays an important role in the biological process of eukaryotes, this kind of PTM is rare and controversial in apicomplexan parasites [14]. However, some reports indicate that *N*-glycosylation is very common in the tachyzoite and cyst stages in *T. gondii* [14,20]. Studies have shown that the *N*-glycans of apicomplexans transmitted by arthropods are very short or absent, while the apicomplexans (i.e., *Toxoplasma*, *Neosporidia*, *Cryptosporidium*, and *Eimeria*) transmitted orally via oocysts have longer *N*-glycan precursors [18]. We speculated that the protein *N*-glycosylation of *T. gondii* is significant and indispensable when it invades its final host intestine. In this manuscript, we identified many known and novel candidate glycoproteins in the feline small intestine following *T. gondii* infection by using LC-MS/MS and further analyzed the pathway involvement of these glycoproteins.

## 2. Materials and Methods

### 2.1. Ethics Approval

The study design was reviewed and approved by the Animal Ethics Committee of Lanzhou Veterinary Research Institute (LVRI), Chinese Academy of Agricultural Sciences (Permit No. LVRIAEC2018-06). The procedures involving animals were carried out in accordance with the Animal Ethics Procedures and Guidelines of the People’s Republic of China. All efforts were made to minimize suffering and to reduce the number of cats used in the experiment.

### 2.2. Laboratory Animals

Kunming mice (8 weeks, 19–20 g, female, Lanzhou, China) were purchased from the laboratory animal center of LVRI, and had free access to sterile food and clean bottled water and were kept within a spacious cage at room temperature (25 ± 2 °C). The mice were intraperitoneally inoculated with *T. gondii* type II Prugniuad tachyzoites, and tissue cysts of the Pru strain were collected from the infected mice brain; then, it was crushed in a mortar, ground, diluted with phosphate buffered saline (PBS), and stored at 4 °C until use. 

Six domestic cats (Felis catus, Chinese Li Hua breed, 7–9 months, Lanzhou, China) were purchased from a local breeder, and were randomly assigned to the experimental group (10_DPI_1, 10_DPI_2, and 10_DPI_3) and control group (Control), with 3 cats in each group. All of the experimental cats were confirmed to be negative for *T. gondii* by using the modified agglutination test (MAT; cut-off: 1:25) [21], and no other major feline diseases (i.e., feline immunodeficiency virus, feline leukemia virus, feline calicivirus, and feline parvovirus) were detected by commercial ELISA kits (Enzo, EastCoast Bio, Beijing, China) according to the manufacturer’s instructions. Each cat in the experimental group was intragastrically challenged with 600 Pru tissue cysts diluted in PBS, while the cats in the control groups were treated with the same amount of PBS. The cat’s intestinal epithelial tissue was scraped by a cell scraper at 10 days post-infection (DPI), flash frozen in liquid nitrogen, and then sent to the company BGI-Shenzhen (Shenzhen, China) for protein glycosylation analysis. Finally, the remaining laboratory animal tissues and *T. gondii* were subjected to sterilizing treatments such as autoclaving.

### 2.3. Confirmation of Infection Model Establishment

Genomic DNA was extracted from the harvested samples (30 mg) using a commercial kit (QIAamp DNA Mini Kit 50T, Cat. No. 1304, Dusseldorf, Germany) according to the manufacturer’s instructions. The PCR assay detection of *T. gondii* was performed using the B1 gene as described previously [22]. Parasites for histological observation were stained with H&E and observed by light microscopy (Olympus, BX41, Tokyo, Japan) [23].

### 2.4. Total Protein Extraction and Digestion

The sample (15–20 mg) was mixed with 5 mm magnetic beads (BeaverBeads™ his-tag protein purification, Solarbio, Beijing, China) and lysis buffer 3 (containing 1 mM PMSF, 2 mM EDTA, and 10 mM dithiothreitol) and centrifuged to obtain the supernatant, and then 10 mM dithiothreitol was added and the sample was incubated in a water bath at 56 °C for 1 h. After adding 55 mM IAM at room temperature (avoiding light incubation for 45 min), 4 times the volume of acetone (−20 °C, 2 h) was added, and the same volume of acetone (−20 °C, 2 h) was added again two to three times until the supernatant was colorless. The precipitate was collected after centrifugation, mixed with 5 mm magnetic beads with the lysis buffer 3, shaken for 2 min, centrifuged, and used for quantification. Quantification of the protein was carried out using the Bradford Kit (Bradford Assay Kit, Bio-Rad, Shanghai, China) [24] and analyzed by SDS–PAGE [25] as described in the manufacturer’s protocol.

A total of 500 μg of the protein solution was taken from each sample, and the enzyme hydrolysis activity was determined according to the protein solution: trypsin enzyme = 40:1 (37 °C, 4 h). A Strata X column was utilized to remove salt and then the enzymatic peptides were subjected to vacuum drying. A small portion of the peptides was used to detect the enzymatic hydrolysis effect using mass spectrometry.

### 2.5. Enrichment of N-glycosylated Peptides and Deglycosylation

Three hundred microliters of qualified peptides were dissolved in 60% ACN and 0.1% FA, and the optimized HILIC method was then used for enrichment and grouping fractionation by HPLC (5 µm, 150 × 4.6 mm, Merck, Beijing, China) [26]. Fractionated product collection started at the 30th min and ended at the 54th min, with one tube per minute. A total of 24 tubes were collected and labeled and then pumped by a freezer pump. Each of the 4 adjacent tubes were reconstituted with a total of 50 µL of 50 mM NH_4_HCO_3_, combined into one tube, and finally divided into 6 fragments. Then, 2.5 µL of PNGase F was added to each fragment, vortexed, mixed well, centrifuged immediately, and incubated overnight at 37 °C. Then, it was frozen and drained.

### 2.6. High-Performance Liquid Chromatography Coupled to Tandem Mass Spectrometry 

The extracted peptide samples were redissolved in mobile phase A (2% ACN, 0.1% FA) and centrifuged at 20,000× *g* for 10 min, and then the supernatant was used for gradient separation by UHPLC (UltiMate 3000, Thermo, Shanghai, China). The separated peptides were ionized by the nanoESI source and transferred to the tandem mass spectrometer Q-Exactive HF (Thermo Fisher Scientific, San Jose, CA, USA) for data-dependent acquisition (DDA) mode detection [27]. Main parameter settings: The MS1 scan range was 350~1500 *m/z*, and the resolution was set to 120,000. The MS2 started at 100 *m*/*z*, and the resolution was 30,000. The ion fragmentation mode was HCD, and the dynamic exclusion time was set to 30 s. The AGC settings: 1E6 for the first level, and 5E4 for the second level.

### 2.7. Enrichment of N-glycosylated Peptides and Deglycosylation

The host proteins/peptides analysis process: The FFPE sample data generated by the high-resolution mass spectrometer were identified using MaxQuant (Andromeda, v.1.5.3.30, http://www.maxquant.org, 7 February 2019) [28]; the fixed modifications were carbamidomethyl (C) and the variable modifications were oxidation (M), acetyl (protein N-term), and Asn- > Asp (N). The filtration was completed with PSM-level FDR <= 1% at the spectrum level and filtered with protein level FDR <= 1% at the protein level, and the database was searched by the UniProt *Felis catus* and *Toxoplasma gondii* (111,780 sequences). At the same time, the modified sites were filtered with 1% FDR to obtain significant modification results. MaxQuant was used to perform the quantitative analysis based on the peak strength, peak area, retention time, and other information regarding the peptide segment related to the primary mass spectrometry, extracting the peptide segment related to the target modification, and conducting a series of statistical analyses and quality controls. The Welch’s *t*-test (*p*-value) was used to test the significance for MaxQuant, and the modification quantification was screened according to a fold change ≥ 2 and a *p*-value ≤ 0.05 as the criteria for a significant difference.

The parasite proteins/peptides analysis process: The identification of *N*-glycosylated proteins was mainly based on the matching of experimental tandem mass spectrometry data with theoretical mass spectrometry data obtained by the database simulation to obtain protein identification results. The protein identification was performed using Mascot software (Mascot 2.3.02, https://mascot-distiller.software.informer.com/, 1 November 2021). The fixed modifications were carbamidomethyl (C), and the variable modifications were oxidation (M), acetyl (protein *N*-term), and Asn -> Asp (N). The filtration was completed with the peptide mass tolerance set at 20 ppm and filtered with the fragment mass tolerance set at 0.05 Da. The max missed cleavages was 2, and the database was searched using the UniProt *Toxoplasma gondii* library (77,541 sequences, https://www.uniprot.org/, 8 January 2019). This process used Percolator (http://noble.gs.washington.edu/proj/percolator, 1 November 2017) to preprocess and re-score the results generated by the search engine to improve the accuracy of the correct matching and random matching. The output results were then filtered with PSM-level FDR <= 0.01 to obtain a list of significantly identified spectra and peptides. Then, based on the “parsimony principle”, this process performed protein inference on the peptides and generated a series of proteomes.

In this study, the Universal Protein Resource (UniProt) database (https://www.uniprot.org/, 1 April 2022) was used to annotate the identified proteins. The Kyoto Encyclopedia of Genes and Genomes (KEGG) Orthology-Based Annotation System 3.0 (KOBAS) (http://bioinfo.org/kobas, 20 December 2020) is a database resource for pathway annotation. KEGG is the main public database of pathways, and pathway analysis can identify the major biochemical metabolic pathways and signal transduction pathways in which proteins are involved [29]. The Gene Ontology (GO, http://geneontology.org/, 7 October 2022) database was used for the gene enrichment analyses, and GO function annotation analysis was performed on all of the identified proteins. The results include protein2go (providing a list of IDs of all corresponding GO functions for each protein) and go2protein (i.e., biological process, BP; cellular component, CC; molecular function, MF). The Clusters of Orthologous Groups of proteins (COGs) database (https://www.ncbi.nlm.nih.gov/research/cog/, 1 March 2022) was used for the orthologous classification of the proteins. In addition, the Minimal Common Oncology Data Elements (MCODE) plugin for Cytoscape software (https://cytoscape.org/, 21 April 2021) was applied to analyze the highly interconnected clusters in the protein–protein interaction (PPI) network using the default parameters.

## 3. Results

### 3.1. Differential Quantitative Analysis of the Modified Peptides/Proteins

After challenge with *T. gondii* (TOX) in the small intestine of cats, the DNA detection of the B1 gene showed that T. gondii-infected cat models were successfully established (Appendix A). The pathological tissue section of the cats’ small intestines also illustrated the successful establishment of infection. As shown in Appendix A, compared with the control group, the experimental group showed swelling of the muscle fibers in the muscle layer (Appendix A). Appendix A shows that the number of intestinal microvilli in the experimental group decreased sharply due to the stromal hyperplasia squeezing the intestinal microvilli. In addition, the experimental group also had basal intestinal hemorrhage accompanied by muscle tissue hyperplasia. The diffuse hemorrhage of the intestinal villi was also found in the infected small intestine (Appendix A). All of these results indicate that the model of *T. gondii* infection in the small intestine of cats was successfully established and could be applied to further study.

In this study, a total of 12,202 proteins, 3122 (Felis: 2656, TOX: 466) modified peptides, and 1813 (Felis: 1411, TOX: 402) modified proteins were identified by analyzing the glycosylation results of the T. gondii-infected small intestine of cats (the details are listed in Appendix A). The mass distribution of the identified modified proteins and the number of unique peptide are shown in Figure 1A,B, and the motif distribution of the posttranslational modification sites of the identified modified peptides is shown in Figure 1C. The Welch’s *t*-test (fold change ≥ 2 and q-value < 0.05) was utilized to identify the differentially glycosylated peptides between the experimental group and the control group. Fifty-eight upregulated and 37 downregulated modified peptides with significant differences were detected in the infected small intestine (Figure 2 and Appendix A). The two most significantly upregulated genes (M3XF94_FELCA and M3W4C1_FELCA) and the two most significantly downregulated genes (A0A2I2U7X3_FELCA and A0A2I2UHR2_FELCA) are shown in Figure 2. In addition, we identified 8 N-glycosylated *T. gondii* proteins including 8 N-glycosylated peptides and 8 N-glycosylation sites (Table 1 and Appendix A). The identification map of N-glycosylation peptides is shown in Appendix A, and the distribution of N-glycosylation motifs is shown in Appendix A. An interesting finding is that the protein A0A086JND6_TOXGO (putative translation elongation factor 2 family protein, eEF2) was identified in groups 10_DPI_1, 10_DPI_2, and 10_DPI_3, and the protein A0A086K8H6_TOXGO (ribosomal protein RPL11) was identified in 10_DPI_2 and 10_DPI_3. Another meaningful discovery was that the peptide sequence NMSVIAHVDHGK (A0A086JND6_TOXGO) was identified in 10_DPI_1 (once), 10_DPI_2 (twice), and 10_DPI_3 (twice), and the peptide sequence KKNFSDSGNFGFGIQEHIDLGIK (A0A086K8H6_TOXGO) was identified once in 10_DPI_2 and 10_DPI_3 (Additional Appendix A). Even more coincidentally, we found that peptide_seq NMSVIAHVDHGK was not only a part of *T. gondii* protein A0A086JND6_TOXGO but also a part of protein A0A139Y075_TOXGO and protein A0A2G8Y563_TOXGO.

### 3.2. Functional Annotation and Enrichment Analysis of the Proteins with Differentially Modified Peptides

A total of 42 GO terms, including 19 biological process (BP) terms, 15 cellular component (CC) terms, and 8 molecular function (MF) terms, were significantly enriched with the proteins corresponding to 95 (58 up- and 37 downregulated) differentially modified proteins in the small intestine of cats (Figure 3). In the BP category, the top two upregulated enriched GO terms were single-organism process and cellular process, and the top two downregulated enriched GO terms were cellular process and metabolic process. In the CC category, membrane and cell/cell part were the top two upregulated GO terms, and the top two downregulated enriched GO terms were cell and cell part. In the MF category, the top two upregulated/downregulated GO terms were catalytic activity and binding. In addition to these common GO terms, some special BP terms (biological regulation and response to stimulus), CC terms (cell junction), and MF terms (signal transducer activity, structural molecule activity, and molecular function regulator) were also enriched. Interestingly, there were also many *T. gondii*-modified proteins involved in these pathways, for example, cellular process and metabolic process, cell and cell part, and catalytic activity.

The 1211 proteins were divided into 24 functional classes, among which “general function prediction only” accounted for 14% (166/1211), “posttranslational modification, protein turnover, and chaperones” (11%, 129/1211), and “cell wall/membrane/envelope biogenesis” (10%, 117/1211) accounted for the top three. We also found that in these host-related COG classifications, some of the *T. gondii* proteins were also involved in eight classifications, but the proteins involved are relatively rare (Figure 4). “General function prediction only” included the WD40 repeat, secreted protein containing bacterial Ig-like domain, and vWFA domain, predicted pyrophosphatase or phosphodiesterase, alkaline phosphatase (AlkP) superfamily, uncharacterized conserved protein YfaS, alpha-2-macroglobulin family, tetratricopeptide repeat, uncharacterized conserved protein RhaS, contains 28 RHS repeats, GTPase SAR1 family domain, DNA-binding beta-propeller fold protein YncE, predicted ATPase, etc. The function of “cell wall/membrane/envelope biogenesis” is numerous and mainly involves membrane proteins involved in colicin uptake, murein tripeptide amidase MpaA, *O*-glycosyl hydrolase, and the PASTA domain, which binds beta-lactams. The details are presented in Appendix A. The “Posttranslational modification, protein turnover, and chaperones”-specific functions included the molecular chaperone DnaK (HSP70 and A0A2G8YAT4_TOXGO), molecular chaperone (HSP90 family and B9QJS4_TOXGV), peptidyl-prolyl cis-trans isomerase–rotamase (cyclophilin family and Q26994_TOXGO), chaperonin groEL (HSP60 family), zn-dependent amino- or carboxypeptidase (M28 family), serine protease inhibitor, asparagine *N*-glycosylation enzyme (membrane subunit Stt3), cysteine protease (C1A family), serine protease (subtilase family), and glutathione s-transferase.

The identified modified peptides in this study were mainly annotated into six major subsystems (i.e., metabolism, genetic information processing, environmental information processing, cellular processes, organismal systems, and human diseases), as shown in Figure 5. The most commonly involved pathways were signal transduction, transport and catabolism, immune system, and infectious diseases. Figure 6 shows the top 30 enriched KEGG pathways of the host, and the top three significantly enriched pathways were African trypanosomiasis, fatty acid elongation, and primary immunodeficiency. Meanwhile, the KEGG enrichment results of the *T. gondii*-related protein are shown in Table 2; among them, A0A086JND6_TOXGO participates in the AMP-activated protein kinase (AMPK) signaling pathway and the oxytocin signaling pathway.

### 3.3. PPI Networks for Differentially Modified Peptides/Proteins

The PPI networks were established of the differentially modified peptides/proteins using the STRING database. We found that a variety of interactions in the host cells were up-/downregulated at the glycosylation level at 10 days after *T. gondii* infection. Then, we applied cytotype MCODE for further analysis of the highly relevant subnetworks and identified three MCODE clusters (i.e., viral myocarditis, fatty acid elongation, and protein processing in endoplasmic reticulum) (Figure 7). These analysis results were consistent with those of the KEGG enrichment analysis. The proteins ICAM1 (M3WBF1_FELCA) and CDH17 (M3WMR7_FELCA) were upregulated, while ITGAL (M3 WC52_FELCA) and ITGAE (M3W8N0_FELCA) were downregulated in the viral myocarditis pathway. In the fatty acid elongation pathway, we found that RPL3 (M3W4R9_ FELCA) was upregulated, while PPT1 (M3X4P7_ FELCA) and PPT2 (M3W2C6_ FELCA) were downregulated. For protein processing in the endoplasmic reticulum pathway, in addition to the downregulation of EDEM3 (M3W7T1_FELCA), the other involved proteins, COPB2 (M3VWP9_FELCA), SEL1L (M3WAN5_FELCA), LMAN2 (M3XF94_FELCA), and ERGIC3 (M3VZZ0_FELCA), were all upregulated.

## 4. Discussion

Glycosylation is an important posttranslational modification. Most membrane proteins and secretory proteins are glycoproteins [30]. Glycosylation modification not only affects the spatial conformation, activity, transportation, and localization of proteins but also plays a critical role in signal transduction, molecular recognition, and immunity [31,32]. In this study, we performed a global glycosylation modification analysis of *T. gondii*-infected small intestines of cats after 10 DPI by using LC-MS/MS technology. A total of 1813 modified proteins (Felis: 1411, TOX: 402) were identified in the glycosylation analysis after *T. gondii* infected the terminal host, indicating that glycosylation of *T. gondii* may occur frequently in the process of infection with definitive hosts, which is consistent with previous reports of the glycosylation of *T. gondii* [14]. A total of 3122 (Felis: 2656, TOX: 466) modified peptides were identified. Meanwhile, under strict screening conditions, we found that there were more upregulated differentially modified peptides (56) than downregulated ones (37), which may mean that most of the cat-modified peptides were upregulated after *T. gondii* infection.

Among these findings, eight *N*-glycosylated proteins and eight *N*-glycopeptides of *T. gondii* were screened. We infer that the eEF2 protein (A0A086JND6_TOXGO) and its corresponding peptide sequence (NMSVIAHVDHGK) may be meaningful in future research. We noted that eEF2 plays a major role in the protein synthesis and survival of *Eimeria tenella* [33], and low levels of total eEF2 protein in *T. gondii* reduced TgPHYb expression [34]. eEF2 has been identified or confirmed as an antimalarial and anticoccidial drug target of *Plasmodium falciparum* and *Eimeria tenella*, respectively [33,34]. It is clear that the eEF2 of *T. gondii* localizes to the nucleus [35]. Based on these findings, we speculate that the eEF2 protein and its corresponding peptide sequences (NMSVIAHVDHGK) may be quite unexpected candidate glycoproteins for *T. gondii* control, and its glycosylation status and contribution to *T. gondii* and host will await further experimental investigations.

The results of the GO analysis of the glycosylated proteins after *T. gondii* infection in cats suggested that several special GO terms, such as biological regulation and response to stimulus, that play important roles in the NF-κB signaling pathway and the regulation of MIC protein were enriched in *T. gondii* and the cats [36,37]. The cell junction term plays an important role in maintaining the invasive force of *T. gondii* [38]. Signal transducer activity, structural molecule activity, and molecular function regulator terms are linked to the process by which ROP16 is released from rhoptries into the host cell [39,40]. The GO terms cellular process and metabolic process, cell and cell part, and catalytic activity were not only involved in the pathway enrichment of the host but were also a significant pathway of *T. gondii*, which indicates that after *T. gondii* infects the host, some pathways changed significantly and synchronously between the host and *T. gondii*.

The COG potential function prediction showed that the identified glycosylated proteins were mainly enriched in three parts: “general function prediction only” (14%); “posttranslational modification, protein turnover, and chaperones” (11%); “cell wall/membrane/envelope biogenesis” (10%); moreover, each of them contained many functional proteins. For example, in the “posttranslational modification, protein turnover, and chaperones” section, the HSP70 and HSP90 families are not only molecular chaperones [41] but also glycosylated proteins of *T. gondii* identified in this study. They are related to *T. gondii* differentiation, which is an important process in *T. gondii* pathology [42]. Taken together, these findings indicate that the glycosylation of Hsp90/Hsp70 could act as a key process in the life of *T. gondii*. The cyclophilins of peptidyl-prolyl cis-trans isomerases (PPIases) are believed to be involved in protein folding [43], and their role in *T. gondii* may be regulated by P glycoprotein [44]. Heat shock protein 60 (TgHSP60) plays an important role in intracellular survival and in the differentiation of *T. gondii*, and it has been isolated from cats [45]. In our study, HSP60 was identified just after the cat was infected with *T. gondii*, which further shows that HSP60 does play an important role in the differentiation of oocysts after cat infection. Serine protease inhibitors display immunomodulatory properties [46], and *T. gondii* serine protease inhibitor 1 (TgPI1) not only affects the virulence of *T. gondii* but also affects the differentiation of bradyzoites [47]. It has been speculated that this may occur through inhibition of *T. gondii* and/or host serine protease [47]. In our study, serine protease inhibitors were detected in the host glycosylated protein after *T. gondii* infection in cats, which also confirmed this inference from another aspect.

The KEGG enrichment results for the glycosylated proteins showed that the signal transduction, transport, and catabolism; immune system; infectious diseases were significantly enriched, indicating that these pathways play an important role in the glycosylation process of the host. In addition, fatty acid elongation and primary immunodeficiency pathways were the top significantly enriched pathways, and these two pathways are closely related to cat glycosylation. For example, acetyl-CoA synthetase is essential for the fatty elongation pathway to generate fatty acids used for *T. gondii* membrane biogenesis [48], and endoplasmic reticulum-localized fatty acid elongation and very long-chain unsaturated fatty acids are essential for *T. gondii* growth but are not supplied by the host cell [49]. In addition, the AMPK signaling pathway and oxytocin signaling pathway were significantly enriched during the process of *T. gondii* infection, and both are involved in the *T. gondii* A0A086K8H6_TOXGO protein. AMPK signaling is required for CD40-induced autophagic killing of *T. gondii* and plays a pivotal role in regulating cellular energy homeostasis [49]. Based on this, we can speculate that the AMPK signaling pathway may significantly change the cell’s metabolism after *T. gondii* infects the definitive host.

Three MCODE_Clusters (i.e., viral myocarditis, fatty acid elongation, and protein processing in endoplasmic reticulum) of Cytoscape by the differentially expressed peptides/proteins are shown in Figure 7. Regarding the viral myocarditis pathway, *T. gondii* can cause severe myocarditis and even death when people have immune deficiency or immune insufficiency [50,51]. Intercellular adhesion molecule 1 (ICAM-1) is not only upregulated on cellular barriers when the host is infected but is also related to *T. gondii*, which exploits the host to cross cellular barriers and disseminate to deep tissues [52]. In this study, ICAM-1 was also upregulated, and we speculated that this upregulation may be linked to the host migration of oocysts. *T. gondii* possesses a single cytosolic acetyl-CoA synthetase (TgACS) that is involved in providing acetyl-CoA for the essential fatty elongation pathway to generate fatty acids used for membrane biogenesis [48], and it was found that *T. gondii* oocyst walls were acid fast and contained long fatty acyl chains that might be synthesized by an abundant polyketide synthase [53]. We assume that this may be because the host deliberately coats the oocyst wall with acid-fast lipids to make them resistant to the environment. *T. gondii* could activate the unfolded protein response upon endoplasmic reticulum stress, and this pathway could not only cause activation of autophagy but also induce phosphorylation of *Toxoplasma gondii* eIF2α and inhibit translation initiation [54,55]. PPT1 of *T. gondii* is based on amino acid residues distal to the palmitoyl cysteine, which has distinct specificity and could be used to detect the activity of a specific depalmitoylase in complex proteomes [56]. All of these highlighted pathways were found in the differentially glycosylated proteins, whether the alteration contributed to the migration or formation of *T. gondii* oocysts in the cat’s small intestine remains to be determined.

## 5. Conclusions

*T. gondii* oocysts cause great harm to humans and the surrounding environment. In this study, we found that *N*-glycosylation occurs in infected cats and *T. gondii*, and the *N*-glycosylation of eEF2 in *T. gondii* may be an important clue for studying the sexual reproduction of *T. gondii*. The results of the COG and GO analyses showed that the “posttranslational modification, protein turnover, and chaperones” pathways were significantly enriched during the glycosylation process of *T. gondii*-infected cats. This proteome data provides a landscape of *N*-glycosylation in the protein of *T. gondii* and the infected cat.

## Figures and Tables

**Figure 1 animals-12-02858-f001:**
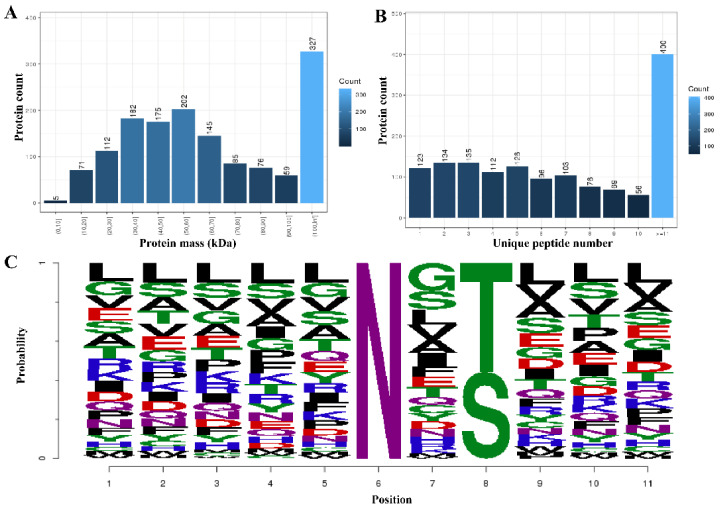
Differential quantitative analysis of the modified peptides/proteins. (**A**) The mass distribution of the modified proteins. The *x*-axis represents the protein amount interval (kilodalton), and the *y*-axis is the number of modified proteins. (**B**) The unique peptide distribution of the target-modified proteins. The *x*-axis represents the number of unique matching peptides per protein, and the *y*-axis represents the number of modified proteins. (**C**) The motif distribution of the posttranslational modification sites of all the modified peptides. The *x*-axis represents the base number, and the *y*-axis represents the corrected score. The higher the base height, the higher the probability of the base appearing in the motif.

**Figure 2 animals-12-02858-f002:**
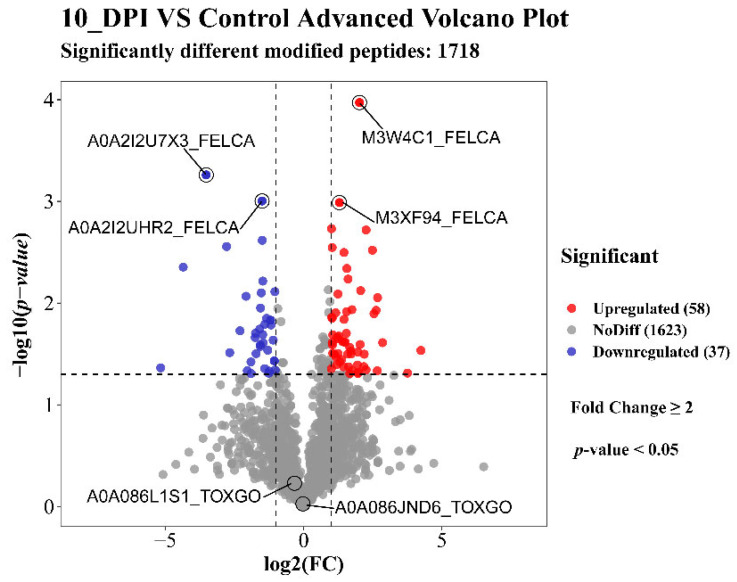
Volcano of the up/downregulated differentially modified peptides. Red indicates upregulation, blue indicates downregulation, and gray indicates no significant difference. The first two significantly enriched upregulated and downregulated modified peptides are marked with horizontal lines. The *Toxoplasma gondii*-modified proteins A0A086JND6_TOXGO (putative translation elongation factor 2 family) and A0A086L1S1_TOXGO (condensin complex subunit 1) were not significantly different.

**Figure 3 animals-12-02858-f003:**
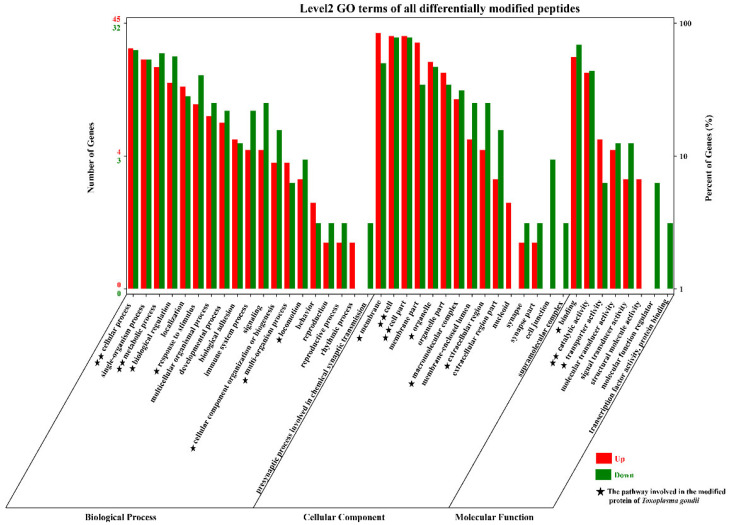
GO term distribution of the proteins sequence corresponding to the modified peptides in the small intestine of cats. Red indicates upregulation, and green indicates downregulation. The bar graph shows the number of modified peptides enriched in GO terms belonging to the three GO categories, biological process (BP), cellular component (CC), and molecular function (MF), at 10 DPI. The *x*-axis represents the GO terms, and the *y*-axis represents the number of upregulated and downregulated modified peptides in the different GO terms. The five-pointed star (★) represents that some *Toxoplasma*-related modified peptides were also involved in the host’s GO pathway. One five-pointed star (★) indicates that less than 4 were involved, and two five-pointed stars (★★) indicate that more than 7 peptides were involved.

**Figure 4 animals-12-02858-f004:**
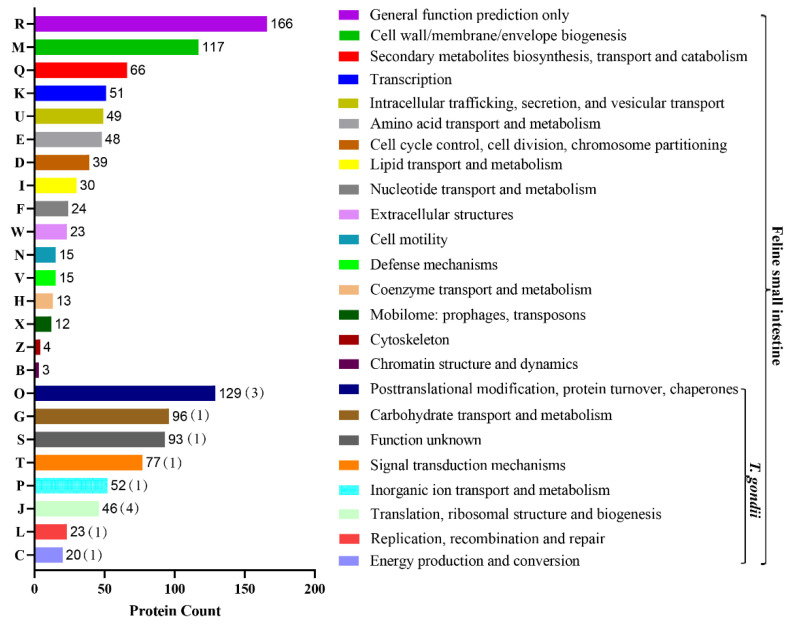
COG term distribution of the modified peptides for feline small intestine. The *x*-axis is the number of proteins corresponding to functional categories, and the *y*-axis is the COG category entries. The graph represents the statistical number of proteins with different functions in the sample. In the COG classification of the 24 host-related modified peptides, 8 (O, G, S, T, P, P, J, L, C)-related modified peptides of *T. gondii* were also involved.

**Figure 5 animals-12-02858-f005:**
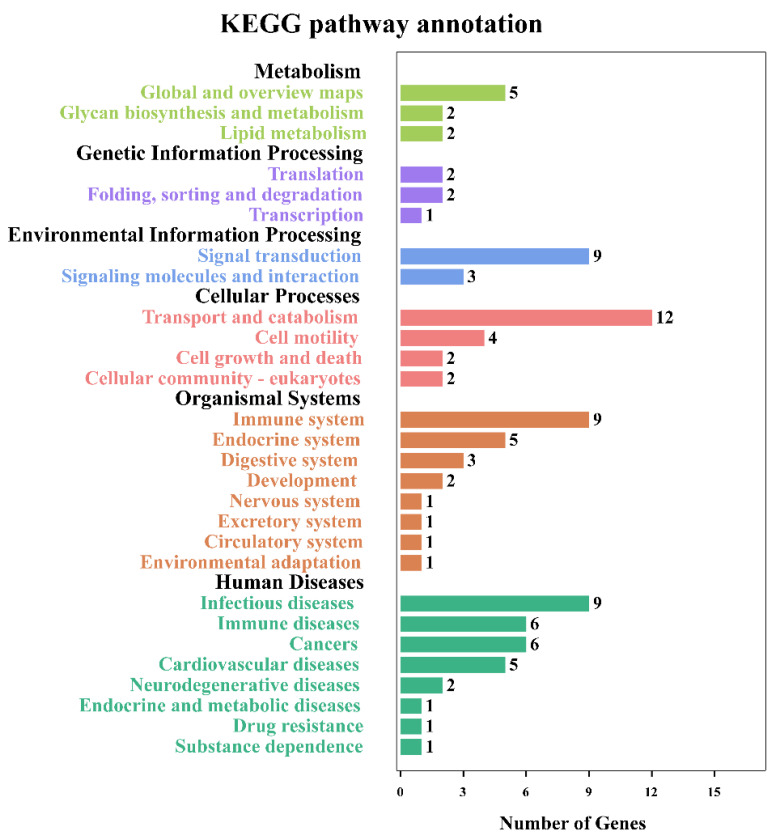
KEGG pathway annotation for the small intestine of cats. The pathways that were significantly enriched for proteins corresponding to the differentially modified peptides. The *x*-axis represents the number of differentially modified peptides in the corresponding KEGG pathways in each KEGG subsystem. The *y*-axis represents the main clusters of the KEGG pathways.

**Figure 6 animals-12-02858-f006:**
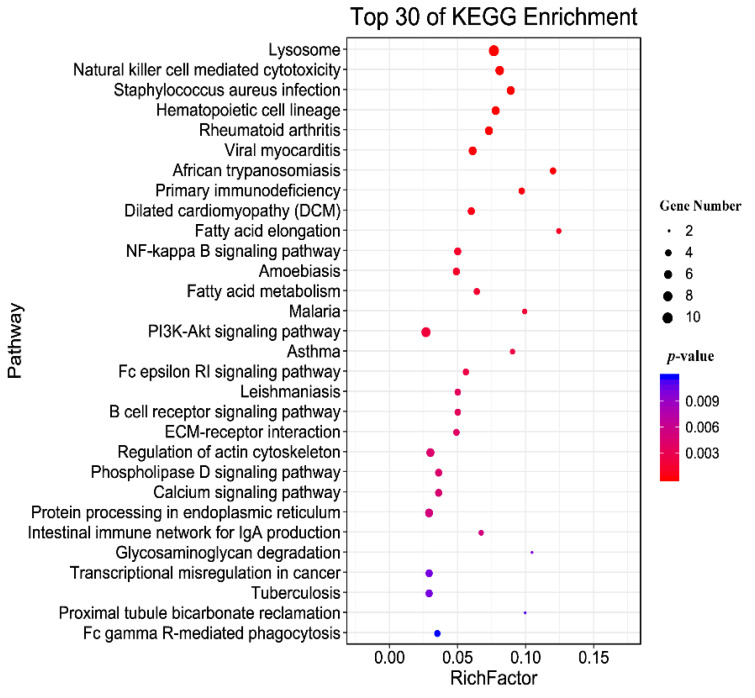
KEGG pathway enrichment of the host. The top 30 enriched KEGG pathways of the identified modified peptides in this study. The *y*-axis represents the distinct KEGG pathways, and the *x*-axis represents the Rich factor. The Rich factor refers to the ratio of modified peptides annotated in the pathway to the total number of modified peptides annotated in the pathway. The greater the Rich factor, the greater the degree of pathway enrichment. The dot size represents the number of modified peptides (bigger dots denote a large number of modified peptides and vice versa). The colors of the dots represent the *p*-values of enrichment: red indicates high enrichment, blue indicates low enrichment.

**Figure 7 animals-12-02858-f007:**
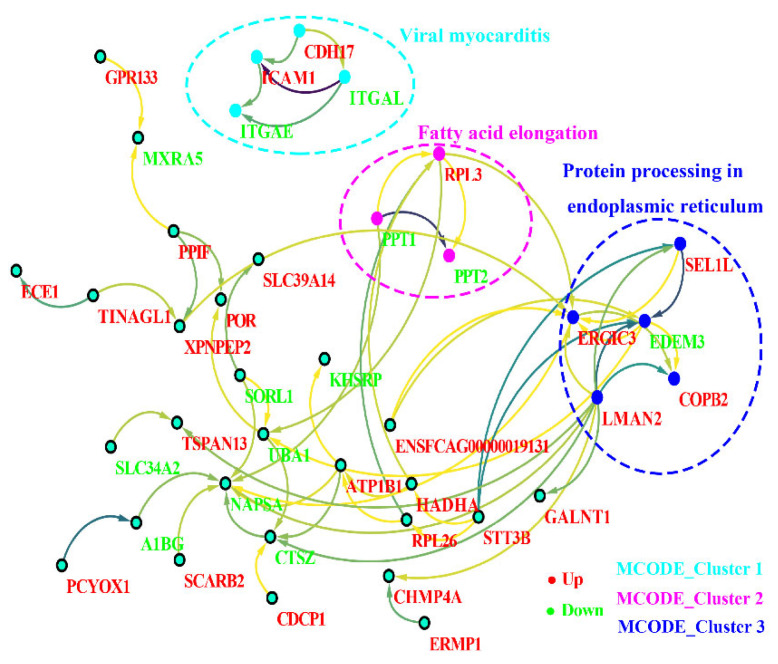
Protein–protein interaction (PPI) networks for the differentially modified peptides/proteins of the host. Red indicates upregulation, and green indicates downregulation. The filled circle represents protein, and the direction of the arrow indicates regulation. The dotted circles represent three MCODE clusters (MCODE clusters 1: viral myocarditis, MCODE clusters 2: fatty acid elongation, and MCODE clusters 3: protein processing in endoplasmic reticulum).

**Table 1 animals-12-02858-t001:** Results of the N-glycosylation identification by *Toxoplasma gondii*.

Sample	Total_Spectra	Identified_Spectra	Proteins	Glycoproteins ^a^	Peptides	Glycopeptides ^b^	Glycosites
10_DPI_1	287,461	1510	209	4	290	4	4
10_DPI_2	288,686	1622	171	4	259	5	4
10_DPI_3	281,373	1491	165	3	240	4	3

^a^ The protein A0A086JND6_TOXGO was identified in 10_DPI_1, 10_DPI_2, and 10_DPI_3, and the protein A0A086K8H6_TOXGO was identified in 10_DPI_2 and 10_DPI_3. ^b^ The peptide_seq NMSVIAHVDHGK was identified in 10_DPI_1 (once), 10_DPI_2 (twice), and 10_DPI_3 (twice), and the peptide_seq KKNFSDSGNFGFGIQEHIDLGIK was identified once in both 10_DPI_2 and 10_DPI_3.

**Table 2 animals-12-02858-t002:** The KEGG enrichment results of the *T. gondii*-related protein.

No.	Pathway	Different Proteins with Pathway Annotation	Proteins	Pathway ID
1	Ribosome	1 (50%)	tr|A0A086K8H6|A0A086K8H6_TOXGO	ko03010
2	AMPK signaling pathway	1 (50%)	tr|A0A086JND6|A0A086JND6_TOXGO	ko04152
3	Oxytocin signaling pathway	1 (50%)	tr|A0A086JND6|A0A086JND6_TOXGO	ko04921
4	Amino sugar and nucleotide sugar metabolism	1 (25%)	tr|A0A151HN22|A0A151HN22_TOXGO	ko00520
5	Aminoacyl-tRNA biosynthesis	1 (25%)	tr|A0A125YXZ8|A0A125YXZ8_TOXGV	ko00970
6	Ribosome biogenesis in eukaryotes	1 (25%)	tr|A0A086KYF0|A0A086KYF0_TOXGO	ko03008

## Data Availability

The mass spectrometry data obtained in this study were deposited in the National Center for Biotechnology Information (NCBI) Sequence Read Archive (SRA) database (https://www.ncbi.nlm.nih.gov/sra), under accession number PRJNA673972, or in Mendeley Data (https://data.mendeley.com/datasets/rn2m56vdky/1).

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
