# Peer review of "Glycosylation Analysis of Feline Small Intestine Following Toxoplasma gondii Infection"

_animals, 2022, doi:10.3390/ani12202858_

Round 1
Reviewer 1 Report
Congratulations, you have done a great job with appropriate experiments and remarkable equipment. I believe your work should be published.
More Comments:
I think the work is relevant because it deals with an infection that is also a zoonosis, which becomes particularly dangerous if it is infected with pregnant women. Furthermore, cats are among the animals responsible for the permanence and resistance of the pathogen in the environment. It is the main animal reservoir of the pathogen. The study conducted by the authors is carried out using very sensitive instruments such as UHPLC connected to a detector as with MS / MS, which serves for the identification of protein sequences. In particular, the study was centered on the analysis of the glycosylation of proteins, which are the basis of various cellular mechanisms. The results are an important element, which helps to shed light on the probable pathways, which are activated and deactivated in the cat host following Toxoplasma infection, therefore they are a probable target to block its propagation cycle.
Author Response
13 Oct, 2022
Dr Ms. Nellie Yu
Editor for
Animals
Dear Ms. Yu,
Re: Manuscript ID: animals-1924777.R1
On behalf of all co-authors, I would like to thank you and the reviewers very much for favorable comments and constructive suggestions on our manuscript (MS) animals-1924777. The reviewers considered our MS to be of general interest to the readership of Animals, and recommended the acceptance of our MS for publication after revisions.
Therefore, we have revised the MS strictly according to the reviewers’ comments and suggestions. We used the “tracked changes” mode in the WORD to show the revised/changed text in the revised MS. Two MS files are uploaded: the “clean version” as “manuscript”, and the one showing “tracked changes” as “supplementary material”. In the following, we detail our point-by-point responses to the comments and suggestions of Reviewer #1. We made all our responses in blue colour for clarity.
Responses to the comments and suggestions of Reviewer #1
Comments and Suggestions for Authors
Congratulations, you have done a great job with appropriate experiments and remarkable equipment. I believe your work should be published.
Responses: We thank Reviewer #1 very much for favorable comments on our manuscript (MS).
More Comments:
I think the work is relevant because it deals with an infection that is also a zoonosis, which becomes particularly dangerous if it is infected with pregnant women. Furthermore, cats are among the animals responsible for the permanence and resistance of the pathogen in the environment. It is the main animal reservoir of the pathogen. The study conducted by the authors is carried out using very sensitive instruments such as UHPLC connected to a detector as with MS / MS, which serves for the identification of protein sequences. In particular, the study was centered on the analysis of the glycosylation of proteins, which are the basis of various cellular mechanisms. The results are an important element, which helps to shed light on the probable pathways, which are activated and deactivated in the cat host following Toxoplasma infection, therefore they are a probable target to block its propagation cycle.
Responses: We thank Reviewer #1 very much for professional and helpful comments on our MS.
We have done our best to address all comments and we sincerely hope that you find our MS revised to your satisfaction. We are looking forward to receiving your editorial decision soon.
With best wishes,
Jiyu Zhang, BVSc, MVSc, PhD
Key Laboratory of Veterinary Pharmaceutical Development,
Lanzhou Institute of Husbandry and Pharma-ceutical Sciences,
Chinese Academy of Agricultural Sciences, Ministry of Agriculture,
Lanzhou, Gansu Province 730050,
The People’s Republic China
Email: zhangjiyu@caas.cn
Jun-Jun He, PhD
College of Veterinary Medicine,
Yunnan Agricultural University,
Kunming, Yunnan Province 650201,
The People’s Republic China
Email: hejunjun617@163.com
Please see the attachment

Reviewer 2 Report
This study presents glycosylation analysis of proteins of the small intestine of feline experimentally infected with Toxoplasma gondii. The study is well designed and provides interesting new protemic data for researchers interested in T. gondii. In general, the manuscript is quite well prepared, but some parts of it need improvements.
The following points should be corrected/considered:
1. Line 128: Please provide a citation referring to the MAT method which was used or describe how the test was conducted.
2. Line 140: Please specify precisely which DNA isolation kit was used.
3. Line 142 Please provide a citation referring to the HE staining that was performed or describe how it was done.
4. Line 145: Please describe in detail which magnetic beads were used.
5. Line 193: Please indicate what software was used to conduct the Welch's t-test.
6. Please explain why MaxQuant was used to identify host proteins while Mascot was used to identify Toxoplasma proteins.
7. Please explain the letters "a" and "b" that are used in superscript in the column headings named glycoproteins and glycopeptides, respectively.
8. Please explain why it was decided to collect samples from cats 10 days after T. godnii infection.
9. Line 455: I think the authors meant "Cytoscape" rather than "Cytotype". Didn't you?
10. Conclusions section: Please describe the conclusions slightly more precisely.
11. Please italicize the Latin names of species throughout the manuscript.
12. Line 521-523: SRA database is not a proteomics data repository. Please explain how proteomic data was deposited in this database?
Author Response
13 Oct, 2022
Dr Ms. Nellie Yu
Editor for
Animals
Dear Ms. Yu,
Re: Manuscript ID: animals-1924777.R1
On behalf of all co-authors, I would like to thank you and the reviewers very much for favorable comments and constructive suggestions on our manuscript (MS) animals-1924777. These comments and suggestions are very valuable for us to revise and improve the quality of our MS. We have revised the MS strictly according to the reviewers’ comments and suggestions. We used the “tracked changes” mode in the MS WORD to show the revised/changed text in the revised MS. Two MS files are uploaded: the “clean version” as “manuscript”, and the one showing “tracked changes” as “supplementary material”. In the following, we detail our point-by-point responses to the reviewer’s comments and suggestions. We made all our responses in blue colour for clarity.
Responses to the comments and suggestions of Reviewer #2
Comments and Suggestions for Authors
This study presents glycosylation analysis of proteins of the small intestine of feline experimentally infected with Toxoplasma gondii. The study is well designed and provides interesting new protemic data for researchers interested in T. gondii. In general, the manuscript is quite well prepared, but some parts of it need improvements.
Responses: We thank Reviewer #2 very much for favorable comments and constructive suggestions on our MS.
More Comments:
The following points should be corrected/considered:
- Line 128: Please provide a citation referring to the MAT method which was used or describe how the test was conducted.
Responses: We thank Reviewer #2 very much for constructive suggestions on our MS. We have provided a citation referring to the MAT method according to your suggestion (line 128).
Key points of MAT operation: Serum samples were tested for antibodies against T. gondii by the modified agglutination test (MAT). Sera were tested at 1:25, 1:50, 1:100 and 1:500 dilutions. Sera with titres of 1:25 or higher were considered positive.
[21] Rani, S.; Cerqueira-Cezar, C.K.; Murata, F.; Sadler, M.; Kwok, O.; Pradhan, A.K.; Hill, D.E.; Urban, J.J.; Dubey, J.P. Toxoplasma gondii tissue cyst formation and density of tissue cysts in shoulders of pigs 7 and 14 days after feeding infected mice tissues. Vet Parasitol. 2019, 269, 13-15.
- Line 140: Please specify precisely which DNA isolation kitwas used.
Responses: We have added the information about DNA isolation kit (QIAamp DNA Mini Kit (50T), Cat.No.1304) on our MS accordingly.
- Line 142 Please provide a citation referring to the HE staining that was performed or describe how it was done.
Responses: We thank Reviewer #2 very much for constructive suggestion on our MS. We have provided a citation referring to the H&E staining on our MS accordingly. Briefly, H&E dyeing steps: pairs of adjacent 5-mm square sections from samples collected at 10 DPI were fixed in 10% neutral buffered formalin solution for 2 weeks, dehydrated in a graded series of ethanol, embedded in paraffin wax, and then cut into 5 μm-thick serial sections on a microtome. They were then stained with hematoxylin and eosin. The reference has been added to the sentence (line 143).
[23] Zhou, C.X.; Zhou, D.H.; Elsheikha, H.M.; Liu, G.X.; Suo, X.; Zhu, X.Q. Global metabolomic profiling of mice brains following experimental infection with the cyst-forming Toxoplasma gondii. Plos One 2015, 10, e139635.
- Line 145: Please describe in detail which magnetic beads were used.
Responses: We thank Reviewer #2 very much for constructive suggestions. We have revised accordingly (lines 145-146). Magnetic beads: BeaverBeads™ his-tag protein purification is a new functional material designed for efficient and rapid purification of amino acid tag proteins. It can directly purify high-purity target proteins from biological samples in one step through magnetic separation. Refer to the following referring for details.
Sun, M.S.; Zhang, J.; Jiang, L.Q.; Pan, Y.X.; Tan, J.Y.; Yu, F.; Guo, L.; Yin, L.; Shen, C.; Shu, H.B.; Liu, Y. TMED2 potentiates cellular ifn responses to DNA viruses by reinforcing MITA dimerization and facilitating its trafficking. Cell Rep. 2018, 25, 3086-3098.
- Line 193: Please indicate what software was used to conduct the Welch's t-test.
Responses: We thank Reviewer #2 very much for constructive suggestion on our MS. Software MaxQuant was used to conduct the Welch’s t-test in this study, and we have added the information on our MS accordingly.
- Please explain why MaxQuant was used to identify host proteins while Mascot was used to identify Toxoplasma
Responses: We thank Reviewer #2 very much for professional and helpful comments on our MS. MaxQuant is a protein identification and quantification software suitable for high-precision mass spectrometry data. We need to quantify the data related to host protein. The relevant parameters are as follows: MaxQuant (1.5.3.30), Enzyme (Trypsin), Minimum peptide length (7), Database (uniprot_felis_catus_and_topolasma_gondii_20190108. fasta (111780 sequences)).
Mascot is an important protein identification software in the field of proteomics. We need to make qualitative analysis of Toxoplasma gondii data. The relevant parameters are as follows: Mascot v2.3, Peptide Mass Tolerance (20 ppm), Fragment Mass Tolerance (0.05 Da), Max Missed Clearages (2), Database (uniprot_toxoplasma_gondii_20190108_nr.fasta (77541 sequences)).
- Please explain the letters "a" and "b" that are used in superscript in the column headings named glycoproteins and glycopeptides, respectively.
Responses: We thank the reviewer very much for helpful comments on our manuscript (MS). a: the protein A0A086JND6_TOXGO has been identified in 10_DPI_1, 10_DPI_2, 10_DPI_3, and the protein A0A086K8H6_TOXGO has been identified in 10_DPI_2 and 10_DPI_3. b: the peptide_seq NMSVIAHVDHGK has been identified in 10_DPI_1 (Once), 10_DPI_2 (twice), 10_DPI_3(twice), and the peptide_seq KKNFSDSGNFGFGIQEHIDLGIK was identified once in both 10_DPI_2 and 10_DPI_3. We have revised the manuscript accordingly (lines 285-289).
- Please explain why it was decided to collect samples from cats 10 days after godniiinfection.
Responses: Dubey and Frenkel found that cats excrete oocysts with a short prepatent period (3~10 days) after ingesting tissue cysts, whereas after they ingested tachyzoites or oocysts, the prepatent period was longer (≥18 days), irrespective of the number of organisms in the inocula (Dubey and Frenkel, 1976; Dubey, 1996, 2001, 2006).
Based on the above background knowledge, in our study, the test animals were artificially fed mouse brain tissues containing cysts (identified by a microscope), and cat small intestine samples were collected on the 10 days after infection. According to time and infection based on the difference between the strains, the cat would exclude oocysts from 3~10 days after infection. So we chose to collect samples on the 10 day after infection, which is more stable and reliable.
- Line 455: I think the authors meant "Cytoscape" rather than "Cytotype". Didn't you?
Responses: We are very sorry for our negligence of the spelling mistakes, “Cytoscape” is correct, and we have modified it (line 473).
- Conclusions section: Please describe the conclusions slightly more precisely.
Responses: We thank Reviewer #2 very much for professional and rigorous advice on our MS. We have improved the MS accordingly (lines 496-503).
- Please italicize the Latin namesof species throughout the manuscript.
Responses: We very much appreciate the careful reading of our MS and valuable suggestions of the Reviewer #2, those comments are all valuable and very helpful for revising and improving our paper. We have checked carefully of all the Latin names and made an extensive modification for our MS (Font marked in blue).
- Line 521-523: SRA database is not a proteomics data repository. Please explain how proteomic data was deposited in this database?
Responses: SRA is the largest publicly-available repository of high throughput sequencing data. The archive accepts data from all branches of life as well as metagenomic and environmental surveys. Our datas were uploaded in September 2020. At that time, i didn't know which database was suitable for uploading our data, so all relevant information (including collection date, sequencing instrument, sequencing method, storage location, etc.) was uploaded. Later, it was found that Mendeley Data (Mendeley Data is a secure cloud based repository where you can store your data, ensuring it is easy to share, access and cite, where you are.) can also store data, so a backup was uploaded in Mendeley Data. We also made an attempt to store our data in Integrated Proteome Resources (iProX, https://www.iprox.cn/page/home.html), but for some reasons it did not succeed, so it was not mentioned in our MS. We will do our best to upload the datas to the database.
We have done our best to address all comments and we sincerely hope that you find our MS revised to your satisfaction. We are looking forward to receiving your editorial decision soon.
With best wishes,
Jiyu Zhang, BVSc, MVSc, PhD
Key Laboratory of Veterinary Pharmaceutical Development,
Lanzhou Institute of Husbandry and Pharma-ceutical Sciences,
Chinese Academy of Agricultural Sciences, Ministry of Agriculture,
Lanzhou, Gansu Province 730050,
The People’s Republic China
Email: zhangjiyu@caas.cn
Jun-Jun He, PhD
College of Veterinary Medicine,
Yunnan Agricultural University,
Kunming, Yunnan Province 650201,
The People’s Republic China
Email: hejunjun617@163.com

Round 2
Reviewer 2 Report
The authors addressed all the points of the review and improved the necessary fragments of the manuscript. I thank the authors for their very precise response to my comments. I have no more suggestions.
In my opinion, the manuscript should be accepted in present form.